# The hierarchical packing of euchromatin domains can be described as multiplicative cascades

**Amra Noa**[1], **Hui-Shun Kuan**[2,3], **Vera Aschmann**[4], **Vasily Zaburdaev**[2,3], **Lennart Hilbert**[1,5]*

**1** Institute of Biological and Chemical Systems, Dept. Biological Information Processing, Karlsruhe Institute of Technology, Eggenstein-Leopoldshafen, Germany, **2** Chair of Mathematics in Life Sciences, Dept. Biology, Friedrich-Alexander University Erlangen-Nuremberg, Erlangen, Germany, **3** Max-Planck-Zentrum für Physik und Medizin, Erlangen, Germany, **4** Master's Program Biology, Faculty for Chemistry and Biosciences, Karlsruhe Institute of Technology, Karlsruhe, Germany, **5** Zoological Institute, Dept. Systems Biology and Bioinformatics, Karlsruhe Institute of Technology, Karlsruhe, Germany

* lennart.hilbert@kit.edu

## Abstract

The genome is packed into the cell nucleus in the form of chromatin. Biochemical approaches have revealed that chromatin is packed within domains, which group into larger domains, and so forth. Such hierarchical packing is equally visible in super-resolution microscopy images of large-scale chromatin organization. While previous work has suggested that chromatin is partitioned into distinct domains via microphase separation, it is unclear how these domains organize into this hierarchical packing. A particular challenge is to find an image analysis approach that fully incorporates such hierarchical packing, so that hypothetical governing mechanisms of euchromatin packing can be compared against the results of such an analysis. Here, we obtain 3D STED super-resolution images from pluripotent zebrafish embryos labeled with improved DNA fluorescence stains, and demonstrate how the hierarchical packing of euchromatin in these images can be described as multiplicative cascades. Multiplicative cascades are an established theoretical concept to describe the placement of ever-smaller structures within bigger structures. Importantly, these cascades can generate artificial image data by applying a single rule again and again, and can be fully specified using only four parameters. Here, we show how the typical patterns of euchromatin organization are reflected in the values of these four parameters. Specifically, we can pinpoint the values required to mimic a microphase-separated state of euchromatin. We suggest that the concept of multiplicative cascades can also be applied to images of other types of chromatin. Here, cascade parameters could serve as test quantities to assess whether microphase separation or other theoretical models accurately reproduce the hierarchical packing of chromatin.

**Data Availability Statement:** All relevant data are within the manuscript and its Supporting information files and at https://zenodo.org/record/4727968#.YIukKT-8q00.

**Funding:** This work is funded via the Helmholtz (www.helmholtz.de) Program Biointerfaces in Technology and Medicine (BIFTM) to LH and AN, Volkswagen Foundation (www.volkswagenstiftung.de) initiative "Life?"' to HSK and VZ, Deutsche Forschungsgemeinschaft (www.dfg.de) priority program "Molecular Mechanisms of Functional Phase Separation" (DFG SPP-2191) to VZ and LH. The funders had no role in study design, data collection and analysis, decision to publish, or preparation of the manuscript.

**Competing interests:** The authors have declared that no competing interests exist.

## Author summary

DNA is stored inside the cell nucleus in the form of chromatin. Chromatin exhibits a striking three-dimensional organization, where small domains group into larger domains, which again group into larger domains, and so forth. While this hierarchical domain packing is obvious from microscopy images, it is still not entirely clear how it is established, or how it should be properly characterized. Here, we demonstrate that multiplicative cascades—a concept from theoretical physics used to characterize for example cloud patterns, galaxy locations, or soil patterns—are also ideally suited to describe the domain-within-domain organization of chromatin. This description is rather simple, using only four numbers, and can thus facilitate testing of competing theories for the packing of chromatin domains.

## Introduction

The packing of a several meters long genome into a cell nucleus of only a few micrometers diameter is often referenced as a remarkable phenomenon. Another striking observation is that, across all investigated length scales, this packing follows an apparent hierarchical organization. Specifically, light and electron microscopy, and more recently biochemical methods in combination with modern DNA sequencing technology, have revealed how small domains of chromatin come together to form larger domains, which in turn come together to form yet bigger domains, and so forth [1–3]. Microscopy approaches have shown hierarchical organization in the chromatin fiber at $\approx$ 10 nm [4, 5], amongst chromatin domains at at the scale of $\approx$ 100 nm [6–9], and also for compartments and territories on the micrometer scale [10]. In the context of biochemical studies, which investigate genome organization via the point-to-point contacts of different positions on chromosomes, the observation of such hierarchical packing has been explained by the fractal globule model [11–13], and more recently by the loop-extrusion model [14, 15]. It is, however, still poorly understood how the hierarchical domain packing as visualized in microscopy images is established.

We have recently addressed the internal patterning of euchromatin in relation to transcription activity, and proposed that a fine-grained pattern of euchromatin domains is established and maintained in line with a microphase-separated state of a microemulsion [16]. During our work with super-resolution images of euchromatin organization, it has become increasingly apparent that euchromatin also exhibits a hierarchical packing of domains. One difficulty in our work was that, to our knowledge, no analysis approaches were in common use that would pay full attention to such a structure. However, for the understanding of euchromatin organization as well as potential underlying microphase separation mechanisms, such an analysis would be highly beneficial. Here, we visualize euchromatin organization in zebrafish embryonic cells by 3D STimulated Emission Depletion (STED) super-resolution microscopy using improved DNA stains, and demonstrate how the euchromatin organization in the images can be described in terms of multiplicative cascades. Multiplicative cascades are a well-established concept from theoretical physics, used to describe how patterns are formed by splitting a large structure into smaller and smaller sub-structures, while consistently adhering to one and the same rule how to execute this splitting. Importantly, a wide variety of complex patterns can be generated based on only four parameters that fully specify the cascade process that leads to the final pattern. Such multiplicative cascades seem ideally suited as a simple description of the hierarchical organization of chromatin domains, as we show here for the example of euchromatin that is organized in line with a microphase separation scenario.

## Results

To assess large-scale euchromatin organization, we recorded 3D STED super-resolution microscopy data from nuclei of pluripotent cells in zebrafish embryos. At the pluripotent stage of development (late blastula), these cells exhibit a euchromatin-only nuclear organization and are distributed throughout the cell cycle [16–18]. Both aspects have proven beneficial for the study of large-scale euchromatin organization by STED super-resolution microscopy in our previous work [16, 19]. To further improve image quality, we assessed the performance of STED-capable fluorogenic DNA stains that recently became commercially available. When we tested the established stain SiR-Hoechst (commercialized as SiR-DNA) in fixed embryos, we found a strong influence of the mounting medium (Fig 1A and 1B), as expected from our previous results [19]. Vectashield almost completely suppressed fluorescence, glycerol allowed for clearly detectable fluorescence, and thiodiethanol-based media (TDE) resulted in almost 10-fold higher fluorescence intensity than glycerol. SiR-DNA in TDE, however, exhibited high levels of cytoplasmic signal, leaving us with reservations towards this stain-media combination. The newly available stain SPY650-DNA also gave high intensities in TDE, with lower cytoplasmic signal. However, when only the STED laser was used, still clearly detectable signal was present, suggesting an unfavorable reexcitation of this stain by the STED laser. Another newly available stain, SPY595-DNA, displayed high levels of nuclear signal in TDE, with the comparatively lowest cytoplasmic and reexcitation signal. (A further stain with comparable application purposes has become commercially available as LIVE 590-DNA during the time of writing [20].) To assess the overall performance of this stain-medium combination, we recorded full volumetric stacks of nuclei with the STED line in full 3D depletion mode. In these stacks, no apparent bleaching occurred and distinct chromatin domains could be seen both in lateral (XY) as well as axial (Z) direction (Fig 1C and 1D). Edge and aberration artifacts that are typical for 3D STED in thick samples could be mostly avoided by careful correction collar adjustment. Considering the preferable evaluation of the SPY595-DNA stain, we proceeded to assess euchromatin organization using this stain in combination with TDE as a mounting medium.

We then assessed what different types of euchromatin organization can be found in the different nuclei in the embryos. An overview scan revealed a diverse range of chromatin organization patterns (Fig 2A). After excluding nuclei of cells preparing for, undergoing, or exiting division, we found a continuum of organization patterns. To illustrate this continuum, we display three typical patterns of organization, which we refer to as smooth, intermediate, and coarse (Fig 2A). Based on our previous work, these typical euchromatin patterns can be related to different states of chromatin in the nucleus [16]. The smooth pattern corresponds to a mixed state, which is found early in the cell cycle when transcription has not yet recommenced. The intermediate pattern corresponds to a microphase-separated state, where euchromatin segregates from RNA-rich regions of the nucleus but remains dispersed into small domains by high levels of transcription. The coarse pattern corresponds to an approach towards a fully phase-separated state, which occurs in cells with lower transcription levels. To more systematically assess these types of euchromatin organization, we quantified how prominent the patterns formed by euchromatin were (image contrast, abbreviated as $C_{DNA}$) and what the characteristic distance is over which euchromatin patterns are formed (correlation length, abbreviated as $L_{corr}$) (Fig 2B). The analyzed example images suggest an inverse relationship, where $L_{corr}$ decreases with increasing $C_{DNA}$. Indeed, an analysis over a wider set of nuclei confirms this inverse relationship (Fig 2C). It appears that the different nuclei form a continuum of euchromatin organization patterns, and the smooth, intermediate, and coarse organization patterns are placed at different positions within this continuum. Such a continuum is in

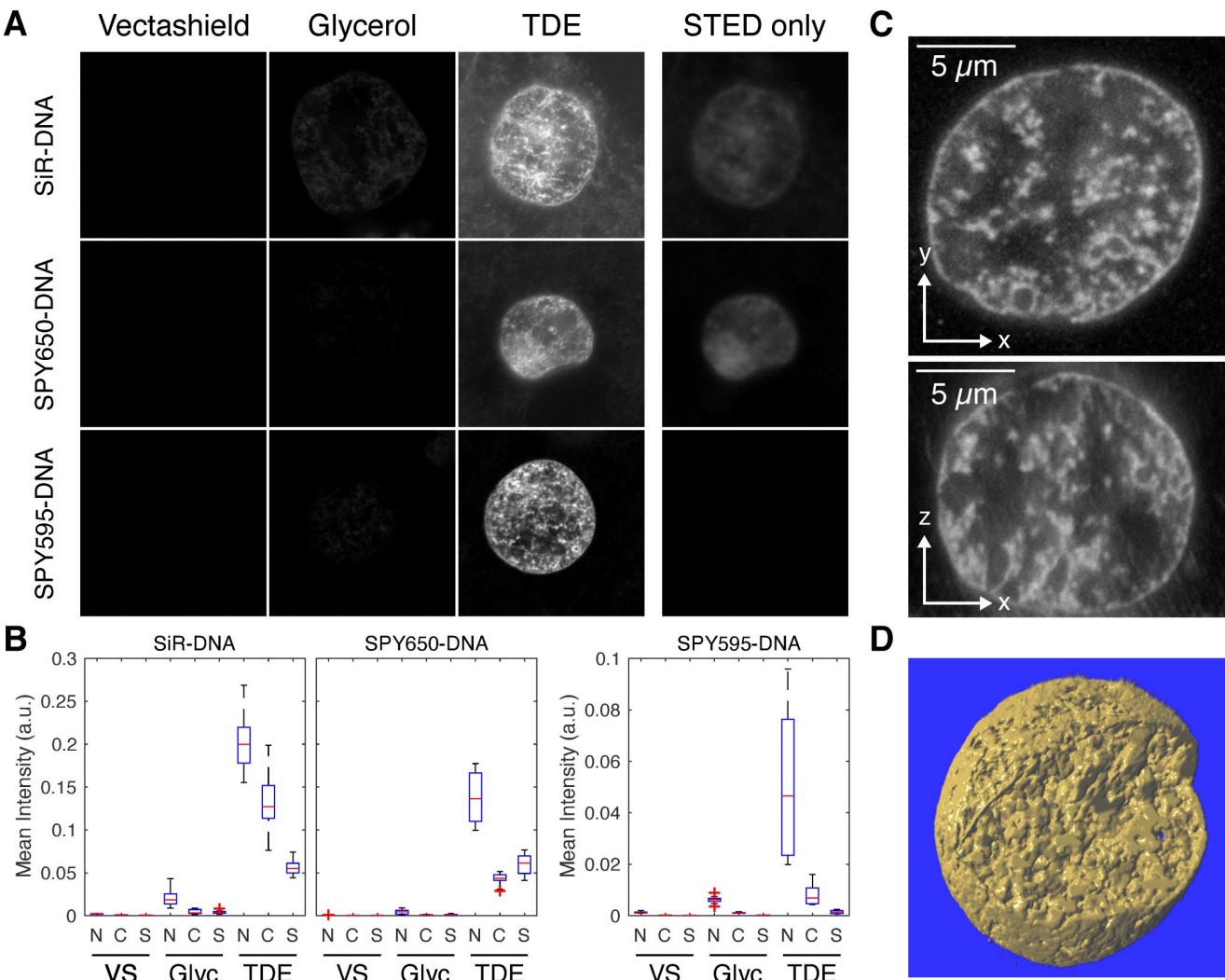

**Fig 1. Improved STED DNA stains allow 3D imaging of entire nuclei and reduction of undesirable signal.** A) Comparison of STED microscopy signal in an older and two improved commercially available DNA stains. All stains were applied in the mounting media Vectashield, glycerol, and thiodiethanol-based media (TDE). SiR-DNA and SPY650-DNA were excited with 652 nm light, and detected in the window from 670 to 740 nm. SPY595-DNA was excited with 599 nm light, and detected in the window from 610 to 640 nm. No gating was applied in detection. In the STED only condition, only the STED laser but no excitation laser was used on the TDE samples. Images were recorded from oblong stage embryos and show a field of view of 25 $\mu$m by 25 $\mu$m B) Quantification of nuclear (N), cytoplasmic (C), and STED only (S) intensity for the different stain-media combinations. Number of assessed nuclei: SiR-DNA $n$ = 5, 10, 5; SPY650-DNA $n$ = 6, 8, 13; SPY595-DNA $n$ = 7, 13, 12. C) XY and XZ slices through a full 3D stack of a nucleus in a SPY595-DNA stained sphere stage embryo. D) 3D rendering of data shown in panel C, one quarter of the image stack was cut away to reveal the inside of the nucleus (Volume Viewer ImageJ plugin was used).

line with previous work, where it was seen that chromatin organization is established progressively after cell division [16, 21]. The observed inverse relationship suggests that the progressive establishment of distinct euchromatin domains (increasing $C_{DNA}$) is correlated with more fine-grained structuring (decreasing $L_{corr}$). Taken together, our analysis reveals an inverse relationship of image contrast and correlation length in our ensemble of nuclei. A packing process that captures the observed euchromatin organization should thus reproduce this relationship.

The observed range of euchromatin organization patterns points towards a packing process that not only results in a domain-within-domain organization, but also allows for a wide range of different patterns. One readily observable natural phenomenon that exhibits (i) a wide

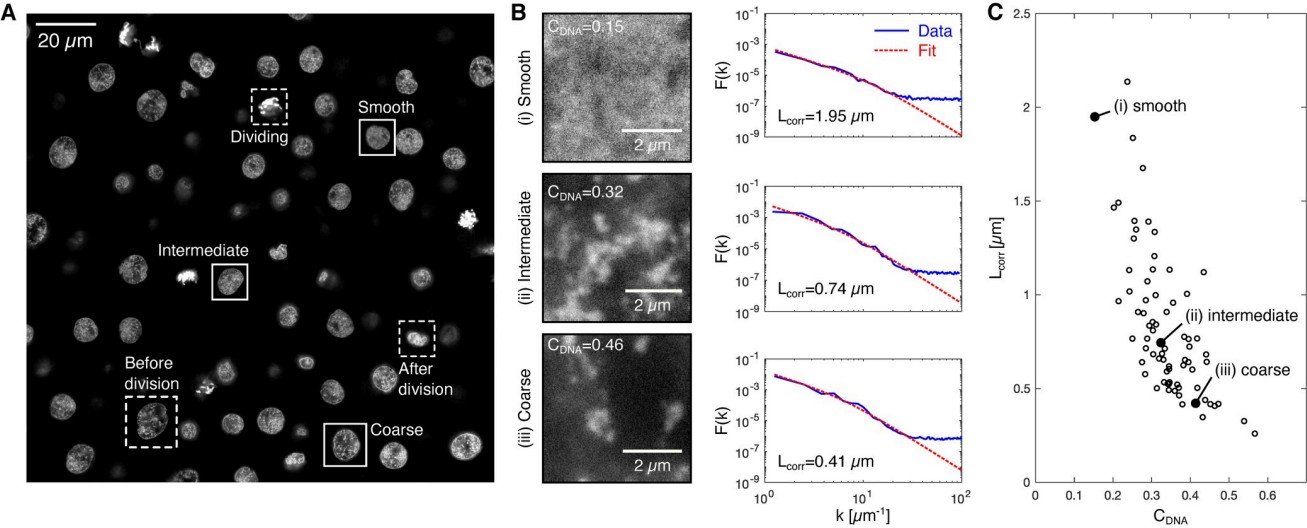

**Fig 2. Cell nuclei in the zebrafish embryo display a continuum of euchromatin organization patterns.** A) Overview of different chromatin organization patterns in nuclei of a sphere stage zebrafish embryo. Examples of different euchromatin organization patterns in interphase nuclei indicated by solid boxes. Examples of nuclei before, during, and after cell division indicated by dashed boxes—nuclei in these cell cycle stages were removed based on visual assessment. Excitation laser 599 nm, detection window 609 to 740 nm, detection gated to within 1.5 to 9 ns after the excitation pulse. B) Representative XY views of image stacks from within nuclei in the smooth, the intermediate, and the coarse euchromatin state. Also shown are image contrast values ($C_{DNA}$), image-based structure factor curves ($F(k)$, see Materials and methods), and the fitted $F(k)$ function used to obtain the correlation length ($L_{corr}$). C) Correlation length vs. image contrast plotted for 78 nuclei, image data obtained from four different embryos.

range of patterns and (ii) placement of domains within increasingly larger domains, are clouds (Fig 3A). Due to the observed placement of ever-smaller replicas of domains into larger domains ("self-similarity"), clouds can be related to fractal sets, in particular to multi-fractals [22–24]. Multi-fractals allow for the recognition of self-similarity in systems with stronger spatial inhomogeneity than encountered in conventional fractal sets. Multi-fractals are commonly characterized using a multi-fractal spectrum, also called an $f - \alpha$ curve (for details, see Materials and methods). Multi-fractal spectra obtained from our microscopy data (S1 Fig) were similar to those seen for other multi-fractal systems [25, 26]. This similarity implies that euchromatin forms patterns with multi-fractal properties. Aiming to not only characterize but reproduce the observed fluorescence intensity distributions, we identified a generative process that is often used to produce multi-fractal patterns, multiplicative cascades [22, 27]. Multiplicative cascades generate density profiles by splitting a square space into ever-smaller sub-squares, while repeating one and the same splitting rule at every repetition (Fig 3B). The splitting rule is defined by four parameter values ($P_1$, $P_2$, $P_3$, $P_4$), which are randomly allocated to the newly generated sub-squares. Note that the splitting rule is stochastic and evaluated using a random number generator as part of the image generation process, so that every generated image is different while nevertheless adhering to the same splitting rule. To test the applicability of multiplicative cascades, we generated an ensemble of density distributions using the multiplicative cascade approach. It was previously suggested that the data collection process should be closely replicated when multiplicative cascades are used to reproduce experimental data [28], so we took into consideration the typical image size, physical pixel size, and effective resolution of our microscope. The resulting density distributions contained cases that, leaving aside the expected traces of the square-based cascade procedure, visually resembled the euchromatin organization patterns observed by microscopy (Fig 3C). Also, the image contrast and correlation length values of some of these examples were comparable to the values obtained for the smooth, intermediate, and coarse patterns in our microscopy data (Fig 3C). When assessing

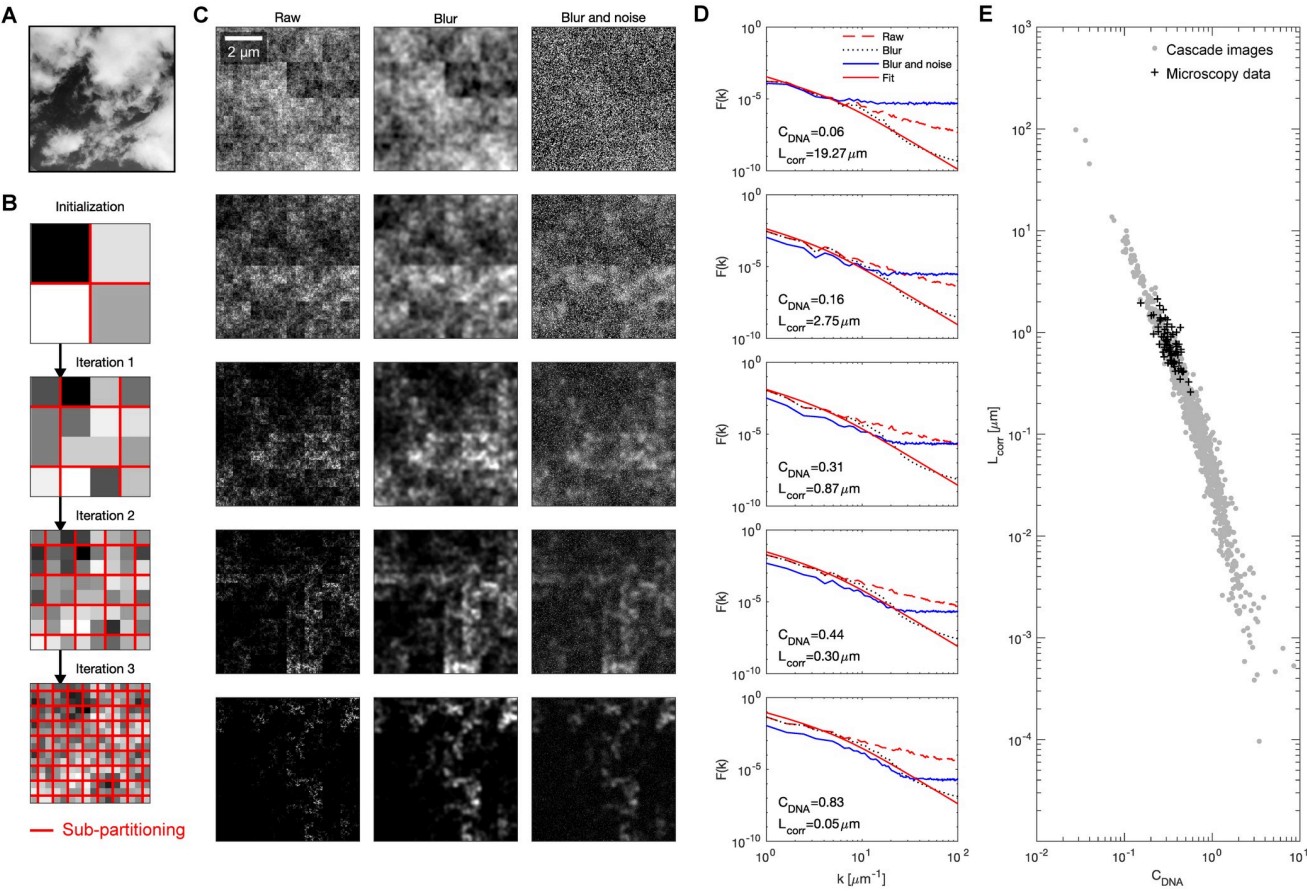

**Fig 3. Images generated using multiplicative cascades show similarities with microscopy images of euchromatin.** A) Example photograph of a cloud. B) Illustration of the image generation based on multiplicative cascades. In the initial 2-by-2 pixel image, the probabilities $P_1$, $P_2$, $P_3$, $P_4$ are randomly assigned. During every further iteration, every pixel gets partitioned into four sub-pixels. As part of this sub-partitioning, each pixel is multiplied with one of the probability values $P_1$, $P_2$, $P_3$, $P_4$, which are randomly allocated. C) Representative examples of concentration profiles generated by multiplicative cascades. Parameter values $P_1$ to $P_4$ used to generate raw distributions of 128-by-128 pixels, from top to bottom: (1, 0.95, 0.9, 0.85), (1, 0.88, 0.76, 0.64), (1, 0.81, 0.62, 0.43), (1, 0.77, 0.54, 0.31), (1, 0.7, 0.4, 0.1). "Blur" images were obtained by smoothing with a two-dimensional Gaussian filter ($\sigma_1$ = 80 nm). "Blur and noise" images were created by further adding Poisson detector noise to every pixel ($\lambda$ = 1, no second blurring step). D) Calculation of $F(k)$ curves and extraction of $L_{corr}$ were carried out identically to microscopy data, polynomial fit and $L_{corr}$ are based on the $F(k)$ curves from the Blur images. E) A set of multiplicative cascades with randomly assigned parameters was executed to allow the comparison of $C_{DNA}$ and $L_{corr}$ values against experimental results.

the whole ensemble of generated patterns, the same inverse relationship of $C_{DNA}$ and $L_{corr}$ as in our experimental data could be seen (Fig 3D). Note, however, that the generated patterns spanned a distinctly wider range of values, suggesting that the observed types of euchromatin organization are contained in a subset of all types of organization that can be generated by multiplicative cascades. This comparison of generated patterns to our microscopy data suggests that multiplicative cascades can indeed reproduce key features of the large-scale organization of euchromatin.

The key property of multiplicative cascades is that the same cascading rule is applied across all length scales. This is often referred to as scale-free organization. Such scale-free organization seems to be in contradiction with the possibility to quantify a specific correlation length. We suspected that the finite resolution of our microscope in combination with the specific patterns of euchromatin organization might be responsible for the observation of an apparent correlation length. In our images generated from multiplicative cascades, we recovered scale-free behavior when we left out the blurring step that approximates the finite microscope

resolution (Fig 3E). When we did apply this blurring step, we saw features indicative of a defined length scale (Fig 3E). This analysis supports a scenario where a multiplicative cascade process, which is inherently scale-free, can reproduce euchromatin organization patterns when the finite microscope resolution is considered.

The cascade-generated images exhibited a close fit to the structure factor curve (see Materials and methods) expected for a microemulsion (Fig 3C), in fact closer than the fit obtained for microscopy data at high spatial frequencies (Fig 2B). High spatial frequencies in microscopic images are prone to be affected by per-pixel detector noise. Indeed, addition of detector noise in the form of a Poissonian emission process to each pixel recovered precisely the deviation from the fit curve at high spatial frequencies that was seen for the microscopy data (Fig 3D).

Now that we established multiplicative cascades as a process that can generate domain-within-domain patterns, we would like to know how the different patterns of euchromatin organization are represented by cascades with specific properties. A given multiplicative cascade process is described by the four parameters, $P_1$, $P_2$, $P_3$, and $P_4$. These parameters are the four probabilities that are used during each of the four-way partitioning steps that make up the multiplicative cascade. To fit these parameters to a given target image, we generated an ensemble of 10,000 candidate density profiles with randomly assigned probability parameters. For the analysis of parameter distributions, we ordered $P_1$ to $P_4$ in descending order throughout. This order does not affect the partitioning process where $P_1$ to $P_4$ are randomly assigned to the four sub-squares. We compared intensity distributions of the generated images resulting from these cascades with the intensity distributions of the target image (Fig 4A and 4B). To this end, the

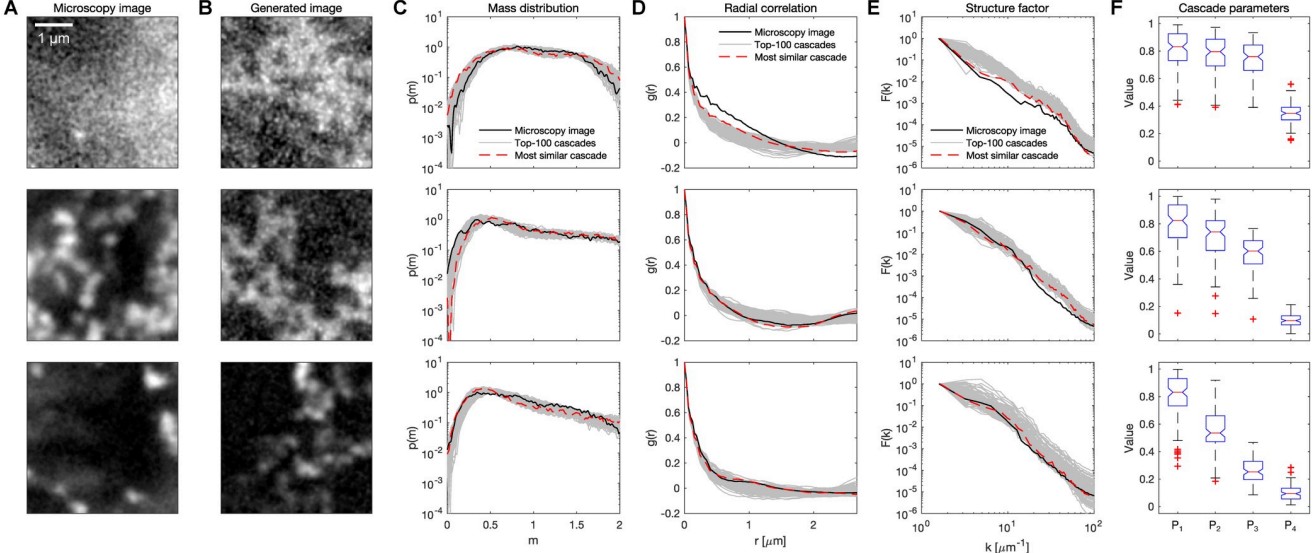

**Fig 4. Multiplicative cascades can be empirically adjusted to reproduce euchromatin patterns observed in microscopy images.** A) Example microscopy images that were used as targets in the adjustment of multiplicative cascades. Intensities are scaled to the range of each image. Raw images were blurred with a Gaussian filter (kernel width 35 nm). B) Generated microscopy images with highest similarity to the microscopy images (64-by-64 pixels, $\sigma_1 = 80$ nm, $\lambda = 1$, $\sigma_2 = 35$ nm, images are from the most similar cascade as chosen by mass distribution and correlation function). C) Mass distributions of the microscopy images, of the most similar generated images, and of the top-100 cascades with highest similarity of mass distribution functions. The top-100 cascades were extracted from an ensemble of 10,000 proposed cascades with randomly assigned parameters. D) Pair-wise radial correlation functions for the microscopy images, the most similar generated images, and the top-100 cascades. E) $F(k)$ curves for the microscopy data, fit of the microemulsion $F(k)$ function, $F(k)$ curve of the raw concentration profiles from the most similar multiplicative cascades, and from the blurred concentration profile from that cascade. F) Box-plots of the cascade parameters of the top-100 cascades. The values $P_1$ to $P_4$ were sorted in descending order before plotting. Note that probabilities are randomly assigned in each sub-partitioning step, so that this sorting did not affect the image generation process.

"top-100" generated images with mass distribution most similar to the target image were chosen (Fig 4C). Without further fitting, the "top-100" generated images also agreed well with the radial correlation function and the structure factor curve of a given target image (Fig 4D and 4E). Out of these "top-100", the generated image with the best fitting correlation function was chosen for further investigation. We verified that this fitting approach can accurately recover known parameters (S2 Fig), so that we can use it for parameter estimation on a larger scale.

Given that the fitted cascades seem to represent the experimental data well, we would now like to interpret the parameter values ($P_1$, $P_2$, $P_3$, $P_4$) associated with the different euchromatin patterns. For all euchromatin patterns, a gradual decrease of the probability values starting from $P_1$ via $P_2$ and $P_3$ towards $P_4$ can be seen (Fig 4F). Note that this decrease is a result of the sorting of the probability values in descending order. Based on the fitting to representative microscopy images, however, this decrease of the parameter values seems to be the more pronounced the more compacted the euchromatin is in the target images (Fig 4F). When we further assessed our entire data set of 78 nuclei, this trend was confirmed as statistically significant (S3 Fig). Lastly, we paid specific attention to the case of intermediate euchromatin compaction, which corresponds to the microphase-separated state of euchromatin organization. In this case, a clear reduction of $P_2$, $P_3$, $P_4$ was seen. However, this reduction was not as pronounced as for the coarse euchromatin state, which is associated with an approach towards full phase separation. This conclusion was also supported by an application of the same cascade fitting approach to simulations of euchromatin microphase separation (S4 Fig) obtained from a lattice model of microemulsification [16]. We conclude that, indeed, specific patterns of euchromatin organization can be represented by multiplicative cascades with particular parameter sets. The microphase-separated state of euchromatin is represented by cascades with a distinct, but not maximally pronounced, decrease of the parameter values $P_2$, $P_3$ and $P_4$.

## Discussion

We demonstrated how the large-scale euchromatin organization in super-resolution microscopy images can be described in terms of the parameters of a multiplicative cascade. Multiplicative cascades are processes that are not defined at a specific length scale or resolution, but rather describe how smaller and smaller parts of a given structure are arranged within bigger parts of the same structure. This concept of a cascading organization seems ideally suited to chromatin organization, where groups of smaller domains come together to form larger structures, which again come together to form yet larger structures, and so forth [1, 4–6, 8, 11–15]. Thus, while we here addressed large-scale euchromatin organization as a specific example, cascade processes in our opinion have potential also for the description of other types of chromatin and other length scales of organization.

A typical zebrafish embryonic nucleus is $\approx 10\ \mu$m in diameter, while our imaging approach can resolve euchromatin organization down to 100 nm or finer. Our investigation thus covers at least two orders of magnitude, at the level of large-scale euchromatin organization. Technical advances in the recent years, specifically in localization microscopy and electron microscopy, have made available yet finer scales of chromatin organization [4, 5, 7–9]. The view of a cascade process, which describes how chromatin domains at smaller levels are hierarchically organized into domains at the next larger levels, might offer a useful language to make sense of the structures revealed by these new techniques. A related image analysis approach recently recognized a similar necessity for the scale-spanning analysis of clustering of nuclear proteins [29]. An interesting question in this context would be whether the same scaling behavior applies across scales, or if the parameters change over the different scales of chromatin organization. Based on previous work, two limits to a truly scale-free packing can already be

identified. First, the scaling behavior changes distinctly at the transition from large-scale chromatin organization to the packing into contact domains [30, 31]. This transition occurs at the scale of a few megabases of sequence length, and at three-dimensional domain sizes of approximately 100 nm [1, 9, 32–36]. Second, going to yet smaller scales of organization, the structure of the chromatin fiber itself is revealed at sequence lengths of a few 100 base pairs, and distances in three-dimensional space of 10 to 30 nm [1, 4, 5]. The chromatin fiber as the fundamental level of genome organization thus represents a hard lower limit on any scale-free organization. The resolution achieved by STED microscopy of DNA in cell nuclei, in practice, ranges from 40 to 100 nm, thus not facing this limit.

The multiplicative cascade description of euchromatin packing simplifies the information contained in a full microscopy image to only four parameters. While this set of parameters is merely a phenomenological description of the domain-in-domain organization of euchromatin, these parameters can be used to decide whether hypothetical mechanisms of euchromatin organization are credible. A hypothetical mechanism that leads to a cascading process with the same parameters as inferred from microscopy images is credible, a process that does not result in the same parameters may be classified as not in line with the data. Such an assessment is distinctly simpler than the comparison against full image data. Considering that microphase separation has been proposed as a mechanism for the organization of different chromatin types and chromatin in general [3, 37–41], such a testing approach that is based in concrete experimental data seems relevant.

The description of euchromatin organization in terms of multiplicative cascades provides a fresh opportunity to test models of molecular access to specific targets. The question of how fast and with what probability for example transcription factors can reach their genomic target sites has been addressed previously [42]. In this context, it was recognized that relatively inaccessible chromatin and the relatively more accessible nuclear space form an interspersed fractal organization, which would directly impact exploration of the intra-nuclear space by various macromolecules [43]. Similarly, multiplicative cascades have previously been used to answer related questions, such as how pressure-injected fluids permeate polymeric networks [44]. The use of multiplicative cascades would allow an abstract treatment of such questions. On the one hand, idealized mesh works generated from cascade processes can be studied with more theoretical rigor and efficiency than real data. On the other hand, predictions from such theoretical work can, based on our fitting to microscopy images, be directly related to specific patterns of euchromatin organization.

Multiplicative cascades produce self-similar structures, which appear the same independently of the length scale they are viewed at. Such structures are often described as fractals. The concept of fractal organization has also been applied successfully to chromatin organization, as studied by microscopy [45, 46], or x-ray scattering experiments [47]. In these and other previous works, the fractal dimension or the strength of self-similarity was assessed. These analysis approaches can be used in combination with the multiplicative cascade approach we suggest here. However, in contrast to these previous approaches, multiplicative cascades do not only characterize experimental data, but can generate synthetic image data without further modifications or assumptions. Further, once the cascading rule is determined, theoretical models that underlie the observed self-similarity can be tested in a much simplified manner.

While our analysis supports the multi-scale, potentially scale-free organization of euchromatin, a microphase separation theory would imply the formation of patterns with characteristic length scales. The question arises how such a theory could still result in the formation of scale-free patterns. One proposed mechanism for microphase separation is the block copolymer nature of the genome, with subregions of different affinities [3, 37]. If the subregions of

such block copolymers would exhibit a scale-free organization in linear space, a scale-free organization in the 3D space filled by such polymers can be intuitively expected. Such scenarios further raise the question how such a scale-free organization of the block copolymers could be brought about. Here, recent theoretical work suggests that local, catalytically self-amplifying chromatin marks can establish microphase patterns [39, 41]. Seeing that self-amplifying processes are frequently implied in the formation of scale-free patterns, such models seem promising candidates.

The overall goal of this study was to capture the hierarchical packing of euchromatin domains in a process that describes not one particular length scale, but rather how the organization at one scale connects to the organization at other scales. We identified multiplicative cascades as an abstract process that can capture the multi-scale chromatin organization seen in 3D STED super-resolution microscopy images by a set of four parameters. When we applied this analysis to euchromatin organization in pluripotent zebrafish embryos, we found that different types of euchromatin organization systematically map to different parameter combinations. In particular, we find that when going from large to smaller scales, chromatin becomes more asymmetrically distributed to sub-regions in patterns that show stronger chromatin compaction. This asymmetry was most pronounced in euchromatin organization that can be explained by phase separation of the euchromatin polymeric chains. The asymmetry in the cascade parameters was still present, but not as pronounced in dispersed euchromatin organization patterns that are associated with a microemulsion or microphase-separated state. These particular sets of parameters provide a well-posed testing scenario for mechanistic models of euchromatin organization, such as microphase separation via microemulsification [16].

## Materials and methods

### Ethics statement

All zebrafish husbandry was performed in accordance with the EU directive 2010/63/EU and German animal protection standards (Tierschutzgesetz §11, Abs. 1, No. 1) and is under supervision of the government of Baden-Württemberg, Regierungspräsidium Karlsruhe, Germany (Aktenzeichen35-9185.64/BH KIT).

### Collection and staining of zebrafish embryos

Late blastula zebrafish embryos were collected and stained as described previously [19]. Briefly, embryos of wild type zebrafish (AB, sourced from the Zebrafish International Resource Center and maintained at the European Zebrafish Resource Center) were obtained by spontaneous mating and dechorionated by pronase treatment. Embryos were collected and fixed (2% formaldehyde, 0.2% Tween-20) at the oblong or sphere stage of development, and subsequently permeabilized (0.5% Triton X-100, 15 minutes at room temperature). Mounting media (Vectashield H-1000, Vector Laboratories, Burlingame, CA, USA; Glycerol; TDE-O, Abberior, Göttingen, Germany) and DNA stains were mixed and applied to the samples on the day of microscopy, as fluorescence intensity decayed noticeably overnight. Fluorescence stains were used at 10 $\mu$M (SiR-DNA, Spirochrome, Stein am Rhein, Switzerland) or 10x recommended dilution (SPY650-DNA, SPY595-DNA, both Spirochrome, molarity not indicated by manufacturer). Embryos were cleaned of yolk particles and placed under #1.5 selected coverslips with spacers.

### Microscopy

Microscopy images were recorded with a Leica TCS SP8 STED microscope (Leica Microsystems, Wetzlar, Germany) equipped with a wavelength-adjustable white light laser, a 775 nm

depletion line, and time-gated HyD detectors, using a motorized-correction 93x NA 1.30 glycerol objective (HC PL APO 93X/1.30 GLYC motCORR). The STED line was used in 100% 3D depletion mode throughout. Within a given experiment and data set, all illumination and acquisition settings were maintained unchanged, so as to obtain quantitatively comparable images. For each acquired image or 3D stack, the correction collar setting was optimized by adjustment for maximal image intensity [48]. For full 3D stacks, the acquisition region was chosen to cover the entire nucleus. For 2D images that were used for $C_{DNA}$ and $L_{corr}$ calculation and for cascade parameter adjustment, a square region was acquired that fit within the nucleus without intersecting the nuclear envelope. The nuclear envelope was excluded to prevent previously observed edge effects [16].

## Image analysis

Staining intensities were quantified using Cell Profiler. Specifically, nuclei were segmented based on the DNA fluorescence intensity using a global Otsu threshold algorithm. The nuclear segmentation mask was dilated to obtain a cytoplasmic mask. For all analyses described below, square images from within a given nucleus were used, making segmentation unnecessary.

   Image contrast ($C_{DNA}$) and correlation length ($L_{corr}$) values were calculated in Matlab. $C_{DNA}$ was calculated as the coefficient of variation of intensity values (microscopy data) or concentration values (generated concentration profiles), dividing the standard deviation by the mean. In the generated data, a correction factor of 0.5 was applied to the $C_{DNA}$ values, to adjust for fluorescence background in the microscopy data [16]. $L_{corr}$ was determined as the pattern length scale above which spatial correlations typical of a microemulsion begin to decay. We determined this length scale $L_{corr}$ by fitting of a functional form expected for the structure factor of a microemulsion. In particular, we derived

$$F(k) = A/(1 + Bk^2 + Ck^4), \tag{1}$$

where $k$ is the spatial frequency in units of $\mu m^{-1}$. This fitting function $F(k)$ relates to the structure factor $S(k)$ as [49]:

$$F(k) = \langle I(\mathbf{k})I(-\mathbf{k})\rangle_k = S(k)V^2\langle I\rangle^2/N. \tag{2}$$

Here $I(\mathbf{k})$ is the two-dimensional spatial Fourier transform of the image intensity values ($I_{i,j}$), $\langle \ldots \rangle_k$ denotes the radial average in the frequency domain ($k = \|\mathbf{k}\|$), $i$ and $j$ the 2D indices in the analyzed plane of the image stack, so that $\langle I \rangle$ is the average intensity. $V$ is the system volume, and $N$ denotes the microscopic number of particles. Here, we assume that image intensity and particle density $N/V$ are proportional. The expression for the fitting function $F(k)$ is taken from microemulsion theory [50]. Specifically, the free energy density is written as $f = a\psi^2 + c_1(\nabla\psi)^2 + c_2(\nabla^2\psi)^2$, using an expansion of the order parameter $\psi$, with $a$, $c_1$, $c_2$ as assignable coefficients. The structure factor $S(k)$ functional form resulting from this free energy density is proportional to $(a + c_1 k^2 + c_2 k^4)^{-1}$, which can be rearranged into the $F(k)$ in our system (Eq (2)). Again, $A$, $B$, $C$ are assignable coefficients. Long-range correlations begin to vanish at a value of $k$ where $Ck^4 = 1$, meaning $k = C^{-1/4}$. Inverting this frequency into a length scale finally gives the correlation length, $L_{corr} = C^{1/4}$.

## Multi-fractal analysis

The multi-fractal features of the images are analyzed with the box counting method [22, 25]. The image is separated into $N(L)$ sub-images with the size $L \times L$, where $L$ has the unit of pixels. The normalized intensity of the sub-image $i$ is $P_i(L) = I_i(L)/I_{tot}$, where $I_i(L)$ is the intensity in the sub-image $i$ and $I_{tot} = \Sigma_i I_i(L)$. A multi-fractal analysis requires the calculation of

generalized moments $\mu_i^q(L)$, which are obtained by raising the normalized intensities to an exponent $q$,

$$\mu_i^q(L) = \frac{(P_i(L))^q}{\sum_i (P_i(L))^q}. \tag{3}$$

Based on the normalized intensities and generalized moments, we can now calculate the generalized fractal dimension, and the $f$ and $\alpha$ values for different exponents $q$, which together make up the $f - \alpha$ curve typically used for multi-fractal analysis. The generalized fractal dimension is defined as

$$D_q = \lim_{L \to 0} \frac{\log \sum_i^{N(L)} [(P_i(L))^q]}{(q-1) \log L}. \tag{4}$$

Further, we calculate

$$f = \lim_{L \to 0} \frac{\sum_i^{N(L)} [\mu_i^q(L) \log \mu_i^q(L)]}{\log L}. \tag{5}$$

and

$$\alpha = \lim_{L \to 0} \frac{\sum_i^{N(L)} [\mu_i^q(L) \log P_i(L)]}{\log L}. \tag{6}$$

to obtain the $f - \alpha$ spectrum. To approximate the limit expressions, we performed linear fits over different $L$ values using the enumerator and denominator as input to the fitting process.

## Generation of concentration profiles by multiplicative cascades

Cascades were carried out based on the four probabilities $P_1$, $P_2$, $P_3$, and $P_4$ as a parameter set. The four probabilities are always sorted in descending order ($1 \geq P_1 \geq P_2 \geq P_3 \geq P_4 \geq 0$). Note that this sorting does not affect the cascade process. The cascades were initialized in a 2-by-2 pixel matrix, and pixels were recursively sub-partitioned assigning the four probability values randomly to the four new pixels (Fig 3B). The recursion was repeated till an image of 256-by-256 pixel size (Fig 3) or 128-by-128 pixel size (Fig 4) was obtained. The concentration in this pixel was then calculated as the product of all probabilities assigned to this pixel during the recursion process. To normalize the concentration profiles, first all pixels were divided by the mean concentration. Then, a pixel size closely matching the pixel size in the microscopy data was assigned (30 nm unless stated otherwise). Generated microscopy images were obtained by applying a Gaussian blur that approximates the limited resolution of the microscopy images (kernel width specified as $\sigma_1$). Noise in the photon detector of the microscope was approximated as a Poisson process by adding exponentially distributed random numbers to every pixel (emission rate specified as $\lambda$). The image was then subjected to another Gaussian blur filter (kernel width $\sigma_2$), as was done for microscopy images. To remove background intensity similarly to microscopy images, the 0.01 percentile intensity value was subtracted from all pixels.

## Estimation of multiplicative cascade parameters

An ensemble of 10,000 concentration profiles was generated, using $P_1$, $P_2$, $P_3$, $P_4$ values randomly drawn from a uniform distribution. For each generated image and each microscopy image, the "mass" present at a given pixel $i$ in that image was calculated, $m_i = I_i / \langle I_i \rangle_i$. Here $I_i$ is the intensity at the pixel $i$. The mismatch between the mass distribution of two images was

then calculated with the error metric

$$\epsilon = \|(\sigma_1)^2 - (\sigma_2)^2\| + \|\gamma_1 - \gamma_2\| + \|\kappa_1 - \kappa_2\|, \tag{7}$$

where $(\sigma_{1,2})^2$ are the variance values, $\gamma_{1,2}$ the skewness values, and $\kappa_{1,2}$ the kurtosis values of the mass distribution $m_i$ of the images 1 and 2. For a given "target" microscopy image, the "top-100" generated images were then chosen based on the lowest values of $\epsilon$. From these top-100 generated images, a single image could further be chosen by the smallest residual error between the radial pair-wise correlation functions of the generated images and the target image (Fig 4).

## Supporting information

**S1 Fig. Euchromatin organization shows multi-fractal characteristics.** A) Example microscopy images that represent the different types of euchromatin organization seen in our data. The displayed images are the same as in Fig 4 of the main article, and are used here for multi-fractal analysis. Scale bar: 1 $\mu$m, note that a larger area of the stacks was used for analysis than in Fig 4. The reference cascade images were obtained by multiplicative cascades based on probability weights $(P_1, P_2, P_3, P_4)$ from previous work [27]. The mono-fractal is obtained using (1, 1, 1, 0), the multi-fractals 1 and 2 are obtained using (1, 1, 0.5, 0.5) and (1, 0.75, 0.5, 0.25), respectively. B) The generalized dimension $D(q)$, and the $f(q)$ and $\alpha(q)$ values for different exponents $q$ are displayed for all images. The values were obtained using the box counting method, using linear fits that were accepted where $R^2 > 0.9$ for all three quantities [22, 25]. For the mono-fractal case, only the dimension $D$ for $q \geq 0$ can be determined numerically by the box counting method; for $\alpha = f = \log(3)/\log(2) \approx 1.585$ the analytically calculated values are indicated [27]. C) $f$–$\alpha$ plots as commonly used to characterize the multi-fractal nature of a given system. A pronounced arc that spans a range of $\alpha$ values is considered indication of multi-fractal patterns. The displayed arcs span values that are similar to previous studies that concluded that systems exhibit multi-fractal patterns [22, 25, 26], as also seen by comparison with the reference cascades. Circles indicate the end points of arcs, as determined by the $R^2 > 0.9$ condition.
(EPS)

**S2 Fig. Fitting cascades via an ensemble of candidate images accurately recovers cascade parameters.** A) Target images of 128-by-128 pixel size were generated using cascades with pre-specified probability parameters $P_1$ to $P_4$ ($\sigma_1 = 80$ nm, $\lambda = 1$, $\sigma_2 = 35$ nm). B) An ensemble of 10,000 candidate images was generated with random probability values, and the best-fitting image was chosen based on mass distribution and radial correlation function similarity. C) Comparison of pre-specified probability parameters (target cascade) and parameters underlying the generation of the best-fitting candidate image (best-fit cascade). Probability values normalized by $P_1$ for better comparability. D) Comparison with distribution of the top-100 candidate images most similar to the target image.
(EPS)

**S3 Fig. Higher contrast in DNA images correlates with more asymmetric multiplicative cascade parameters.** Gray circles are medians of the top-100 parameter sets (evaluated based on variance, skewness, kurtosis comparison). Linear fits (red line) are shown. The Pearson correlation coefficient ($\rho$) and the p value for the particular values of $\rho$ are indicated for each cascade parameter.
(EPS)

**S4 Fig. Fitting of multiplicative cascades to simulated euchromatin distributions reproduces correlations seen in fitting DNA image data.** Synthetic image data from microemulsion simulations of euchromatin organization were used to verify the correlation of multiplicative cascade parameters with image contrast. Simulations for a microemulsified state of euchromatin (untreated case, 120 simulations), a well-mixed state of euchromatin (no transcription, 45 simulations), and the phase-separating case (flavopiridol treatment, 154 simulations) from our previous work were pooled [16]. Image generation parameters for microemulsion simulations: 100-by-100 simulation lattice, pixel size 100 nm, $\sigma_1$ = 200 nm, $\lambda$ = 0.5, $\sigma_2$ = 50 nm, central 32-by-32 pixel region of image used in mass distribution calculation. Image generation parameters for cascade fitting: 32-by-32 pixel images generated by four cascade recursions, pixel size 100 nm, $\sigma_1$ = 200 nm, $\lambda$ = 0.4, $\sigma_2$ = 50 nm. Gray circles are medians of the top-100 parameter sets (evaluated based on variance, skewness, kurtosis comparison). Linear fits (red line) are shown. The Pearson correlation coefficient ($\rho$) and the p value for the particular values of $\rho$ are indicated for each cascade parameter.
(EPS)

## Acknowledgments

Images were acquired at the Karlsruhe Center for Optics and Photonics (KCOP). The fluorescence dyes SiR-DNA, SPY650-DNA, and SPY595-DNA were provided free of charge by Spirochrome.

## Author Contributions

**Conceptualization:** Lennart Hilbert.

**Investigation:** Amra Noa, Hui-Shun Kuan, Vera Aschmann, Lennart Hilbert.

**Supervision:** Vasily Zaburdaev, Lennart Hilbert.

**Writing – original draft:** Lennart Hilbert.

**Writing – review & editing:** Vasily Zaburdaev, Lennart Hilbert.

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
