## [Decision Letter · Decision Letter 0]

3 Feb 2021

Dear Prof. Hilbert,

Thank you very much for submitting your manuscript "The domain-within-domain packing of euchromatin can be described as multiplicative cascades" for consideration at PLOS Computational Biology.

As with all papers reviewed by the journal, your manuscript was reviewed by members of the editorial board and by several independent reviewers. In light of the reviews (below this email), we would like to invite the resubmission of a significantly-revised version that takes into account the reviewers' comments.

We cannot make any decision about publication until we have seen the revised manuscript and your response to the reviewers' comments. Your revised manuscript is also likely to be sent to reviewers for further evaluation.

Sincerely,

Ilya Ioshikhes

Deputy Editor

PLOS Computational Biology

William Noble

Deputy Editor

PLOS Computational Biology

Reviewer's Responses to Questions

**Comments to the Authors:**

Reviewer #1: The article provides an interesting application of a theoretical concept of multiplicative cascades to describe the processes governing euchromatin packing. This theoretical framework has the potential to be used to test mechanistic models of euchromatin organization.

Major revisions:

The article needs to go into further detail of what the descriptive parameters P1, P2, P3, P4 mean in the context of generating the multiplicative cascade images. There is needs to be additional methodological discussion of why the decrease of the probability values from P1 to P4 is common to all patterns of euchromatin observed in the microscopy images.

The authors discuss testing mechanistic models for euchromatin organization such as microemulsification, however they do not provide any direct examples of how multiplicative cascades can be applied to test them. Since the authors already utilize microemulsion theory fitting functions in their methodology, it would be extremely valuable for the readers to see the utility of multiplicative cascades as a test in verifying this model.

The process for generating concentration profiles by multiplicative cascades is done through randomly assigning one of the 4 parameters to each pixel. It would be useful to mention how rerunning the concentration profiles using the same set of parameters affects the resulting images, particularly in the context of their ability to reproduce euchromatin patterns as seen in Figure 4.

Additional comments:

The article should be carefully reviewed for typos. Specifically, in the methods on page 13, (0 ≤ P1, P2, P3, P4 ≤ 1) appears to be the reverse of the decrease of the probability values as described in the results and figures. The word organzation is misspelled in the discussion on page 13.

Reviewer #2: Summary:

This paper presents a new method to model super-resolution microscopy of euchromatin folding within the nucleus, based on a theory of hierarchical chromatin folding using multiplicative cascades. Using this method, the authors determine specific values of 4 parameters required for the multiplicative cascades that can be applied to mimic microphase separation observed in images of euchromatin, which may be useful for testing whether other types of theoretical models accurately reproduce chromatin folding, or for comparisons to future experimental data. In addition, this paper presents an optimized DNA stain/mounting media combination for STED microscopy of nuclear structure in zebrafish embryos. The computational method as applied to chromosome folding super-resolution imaging is original, innovative, and is of importance to researchers in the field of chromosome folding, as it may facilitate comparisons between imaging data and modeling data generated by other means by using the identified multiplicative cascade parameters as a measure of similarity between real and modeled data.

However, there are some conclusions that require additional analysis to be fully supported. In addition, some parts of the methodology are not adequately described or explained, and the underlying computational analyses, raw imaging data and modeling results do not seem to have been made available, and should be included in the final publication.

Major Concerns:

On Page 7, the authors state that “euchromatin intensity distributions showed properties typical of multi-fractals rather than conventional fractal sets”, however Supplementary Material Figure 5 includes only the multi-fractal analysis on the chromatin images, without controls to support this statement. This point would be better supported by also including both a conventional fractal analysis showing that this is not supported by the data, and a comparison to a known multifractal dataset such as the clouds described in the example, or the referenced soil dataset (Adolfo N. D. Posadas, Daniel Gim ´enez, Roberto Quiroz, and Richard Protz. Multifractal Characterization of Soil Pore Systems. Soil Science Society of America Journal, 67(5):1361–1369, 2003.).

On Page 7, the authors state “Image contrast and correlation length values of these examples were also comparable to the values obtained for the smooth, intermediate, and coarse patterns in our microscopy data”, However, in Figure 3B the Lcorr values shown for modeled images that are visually similar to the real images have a much wider range than the Lcorr from the corresponding real images. It would be useful to also show the images corresponding to extremes of Lcorr and Cdna from the model, and representative images that more closely match both the Lcorr and Cdna values. In addition, the authors should add a description of the criteria that were used to choose these examples – were they chosen visually to be similar to the images in Figure 2, by Lcorr and Cdna value, or by Mass Distribution and Radial Correlation similarity as in figure 4?

In Figure 4 A and B, the top generated image does not look very similar to the top microscopy image, however the metrics used to compare them in the rest of the figure look similar. This suggests that some additional metric may be required to be able to robustly compare between real and modeled data, as visually these two images look quite different, and calls into question the later analysis of cascade parameters across all nuclei, as it is not clear how well the modeled images are matched to the real images. A more comprehensive analysis of the matching of modeled and real data would improve this point.

On Page 9, the authors state that “For all euchromatin patterns, a gradual decrease of the probability values starting from P1 via P2 and P3 towards P4 can be seen”, however, it is unclear how P1 to P4 are ordered – are they sorted randomly/based on which was picked first-fourth, or are they sorted based on decreasing value, in which case this would be the only possible outcome? This should be clarified in the methods. In addition, if the ordering is important, are there any examples of images where the values increase or have no trend from P1 to P4, and how do these images look compared to real data? The conclusion that there is more of a difference between the probabilities in more compact states is supported, but I am not sure that the order itself matters based on these results.

On Page 9, the authors state that “Now that we established multiplicative cascades as a process that can generate domain-within-domain patterns, we would like to know how the different patterns of euchromatin organization are represented by cascades with specific properties”. How has the domain-within-domain pattern itself been tested? This is not clear from the text.

On Pages 9 and 11, the finite scale of the microscope vs. the scale-free characteristic of multiplicative cascades is discussed, however it should be noted that in addition to the finite scale of the microscope, the chromatin is also constrained by the size of nucleosomes and other types of DNA binding proteins, and even the persistence length of DNA, and cannot in reality remain scale-free at very small distances, even if the microscopy was not resolution limited. This may affect the conclusions about the applicability of the multiplicative cascade to studying chromatin structure at certain size ranges.

In Figures 2 and 3: It would be useful to add an explanation of why the F(k) fits and data diverge above 10^1.2 for the real data but not for the modeled images. The model generated images seem to be close to the F(k) fit for a much broader range than the real data – is this expected, and how does this affect the results? How is this related to the result in Figure 4 comparing the blurred vs unblurred simulated images?

Minor Comments:

For Figure 2, please add a description of the method that was used to classify the nuclei into interphase vs. cell division, and into the different patterns of euchromatin organization.

In Figure 3: As noted in the text on page 7, In C, Both Lcorr and CDNA for the model have a much wider range than the real data – please expand on the significance of this.

In all figures, it would be useful to note what % of the nuclear area is included in each of these analyzed images, and to note whether multiple areas from one nucleus always show the same type of structure, or can multiple types of structure be found in one nucleus? How does this compare for the modeled images?

Additional protocols/algorithms that are not included, and would be beneficial to share: CellProfiler pipelines used for analysis, Matlab scripts used for analysis, software used for modeling images using multiplicative cascade.

Original imaging data and results from modeling are not available. Something like https://www.ebi.ac.uk/bioimage-archive/ or https://figshare.com/ or https://idr.openmicroscopy.org/ might be appropriate.

Descriptions of Lcorr calculation and the multi-fractal analysis method should be more detailed, some of the variables in the equation seem to be not defined or explained in the text of the methods.

Funder website URLs are not included in the financial disclosure statement, and role of funders in the study is not included in the financial disclosure statement.

Include n for imaging experiment quantifications in Figure 1.

In the title, “The domain-within-domain packing of euchromatin can be described as multiplicative cascades”, the “domain-within-domain” part is not fully supported by the current version of the paper.

**Have all data underlying the figures and results presented in the manuscript been provided?**

Reviewer #1: Yes

Reviewer #2: **No: **CellProfiler pipelines used for analysis, Matlab scripts used for analysis, software used for modeling images using multiplicative cascade, original imaging data and results from modeling should be made available.

PLOS authors have the option to publish the peer review history of their article (what does this mean?). If published, this will include your full peer review and any attached files.

Reviewer #1: No

Reviewer #2: **Yes: **Erica M Hildebrand
---

## [Decision Letter · Decision Letter 1]

16 Apr 2021

Dear Prof. Hilbert,

We are pleased to inform you that your manuscript 'The hierarchical packing of euchromatin domains can be described as multiplicative cascades' has been provisionally accepted for publication in PLOS Computational Biology.

Best regards,

Ilya Ioshikhes

Deputy Editor

PLOS Computational Biology

William Noble

Deputy Editor

PLOS Computational Biology

Reviewer's Responses to Questions

**Comments to the Authors:**

Reviewer #1: The authors have adequately addressed all reviewer comments.

Reviewer #2: The revised manuscript well addresses all of the concerns in my initial review.

**Have the authors made all data and (if applicable) computational code underlying the findings in their manuscript fully available?**

Reviewer #1: Yes

Reviewer #2: Yes

PLOS authors have the option to publish the peer review history of their article (what does this mean?). If published, this will include your full peer review and any attached files.

Reviewer #1: No

Reviewer #2: **Yes: **Erica M Hildebrand

---

## [Editor Report · Acceptance letter]

30 Apr 2021

PCOMPBIOL-D-20-02010R1 

The hierarchical packing of euchromatin domains can be described as multiplicative cascades

Dear Dr Hilbert,

I am pleased to inform you that your manuscript has been formally accepted for publication in PLOS Computational Biology. Your manuscript is now with our production department and you will be notified of the publication date in due course.

With kind regards,

Andrea Szabo
